# Use of Publication Dynamics to Distinguish Cancer Genes and Bystander Genes

**DOI:** 10.3390/genes13071105

**Published:** 2022-06-21

**Authors:** László Bányai, Mária Trexler, László Patthy

**Affiliations:** Institute of Enzymology, Research Centre for Natural Sciences, Eötvös Loránd Research Network (ELKH), 1117 Budapest, Hungary; banyai.laszlo@ttk.mta.hu (L.B.); trexler.maria@ttk.mta.hu (M.T.)

**Keywords:** bystander gene, cancer gene, confirmation bias, funding bias, passenger gene, publication bias, tumor essential gene

## Abstract

de Magalhães has shown recently that most human genes have several papers in PubMed mentioning cancer, leading the author to suggest that every gene is associated with cancer, a conclusion that contradicts the widely held view that cancer is driven by a limited number of cancer genes, whereas the majority of genes are just bystanders in carcinogenesis. We have analyzed PubMed to decide whether publication metrics supports the distinction of bystander genes and cancer genes. The dynamics of publications on known cancer genes followed a similar pattern: seminal discoveries triggered a burst of cancer-related publications that validated and expanded the discovery, resulting in a rise both in the number and proportion of cancer-related publications on that gene. The dynamics of publications on bystander genes was markedly different. Although there is a slow but continuous time-dependent rise in the proportion of papers mentioning cancer, this phenomenon just reflects the increasing publication bias that favors cancer research. Despite this bias, the proportion of cancer papers on bystander genes remains low. Here, we show that the distinctive publication dynamics of cancer genes and bystander genes may be used for the identification of cancer genes.

## 1. Introduction

In a recent study, de Magalhães presented the results of an analysis of PubMed publications on human genes and the ones that also mention the term “cancer” [1]. The author has shown that of the 17,371 human genes with at least one paper in PubMed, 15,233 (87.7%) also have at least one paper mentioning cancer. The overall conclusion of de Magalhães (conveyed by the title of his paper) is that “every gene can (and possibly will) be associated with cancer” and that “if a gene has not been associated with cancer yet, it probably means it has not been studied enough and will most likely be associated with cancer in the future” [1]. The author is correct in pointing out that this conclusion would have important implications for analyzing and interpreting large-scale analyses in cancer genomics, especially as it contradicts the dominant view that cancer is driven by a few hundred cancer genes, whereas the vast majority of genes are just bystanders (or passengers) in carcinogenesis (ICGC/TCGA Pan-Cancer Analysis of Whole Genomes Consortium, 2020; [2,3,4]).

de Magalhães is aware of the fact that since cancer is the most widely studied topic in biological and biomedical sciences, ”human biases can confound systematic analyses” based on publication metrics [1]. Since cancer is one of the most common diseases in modern times, publication bias and funding bias are likely to have a major impact on the number and proportion of publications mentioning cancer. It is important to take this bias into account, as in scientific research human biases (any trend or deviation from the truth in data collection, data analysis, interpretation and publication) can cause false conclusions [5,6].

Although de Magalhães promises to ”explore the extent of such biases in cancer associations and their implications”, these promises remain unfulfilled [1]. Neither the title of the paper nor the abstract has any indication that the author’s intention was to caution readers that the bias might be significant enough to affect the validity of the conclusion drawn. In the last section of the paper, the author emphasizes that ”understanding the reasons for biases in large-scale analyses and correcting for them is of growing importance to increase the value of insights and predictions”, but we find no information about the extent and impact of these biases or how the errors introduced by the biases should be corrected.

Judging from the afterlife of the paper, it appears that its cautionary message has been lost to the readers. The publications referring to the paper cite only its conclusion that every gene is a relevant cancer gene [7,8], leading the citing authors to realize that—if valid—this conclusion would undermine the original mission of The Cancer Genome Atlas (TCGA): to identify a few shared cancer genes.

In cancer genomics, genes are usually assigned to three categories with respect to their role in carcinogenesis: (1) Bystander genes, or passenger genes with no significant role in carcinogenesis; (2) Cancer genes that drive carcinogenesis when they acquire driver mutations; or (3) Tumor essential genes, with functions essential for the growth and survival of tumor cells [3].

The existence of these three distinct categories is supported by the observation that the evolutionary behavior of genes belonging to these categories is markedly different during tumor evolution. For example, the mutation patterns of selectively neutral bystander genes reflect the lack of positive and negative selection, as cancer driver genes show strong signs of positive selection for mutations that drive carcinogenesis, whereas in the case of genes essential for tumor growth, purifying selection dominates [3,9,10,11,12,13,14,15].

In the present study, we have analyzed PubMed data to test whether publication metrics can be used to prove or disprove the critical involvement of a gene in carcinogenesis. The rationale of our approach was that, since science is a self-organizing and evolving network of scholars, projects, papers and ideas, research efforts and seminal discoveries are expected to have a predictable impact on publication metrics [16].

As pointed out by Fortunato and coworkers [16], when scientists select research problems to work on, their choices are shaped by the dilemma of ”productive tradition” or “risky innovation”. Novel findings that appear to be key discoveries are likely to attract numerous research groups to explore the validity and significance of that finding. If the work of these research groups validates and elaborates the novel finding as a seminal discovery, then this is reflected in a significant surge in the number of papers on the given research topic as well as high citation counts of the papers that resulted in a break-through. Conversely, if the research fails to verify the putative significance of the original finding, no burst of publications will follow as negative results are rarely published. As a corollary, validated knowledge is expected to have a characteristic and lasting impact on the dynamics of publication metrics.

In the present work we have monitored how the number and proportion of papers in which the name of genes and the term ’cancer’ co-occur changes as a result of research efforts. Our analyses have confirmed that in the case of the best-known cancer genes, key discoveries triggered a burst of cancer-related publications leading to a sharp and lasting increase in the proportion of PubMed papers on the gene that also mention cancer. In the case of bystander genes, however, the proportion of cancer-related papers remains significantly lower than in the case of cancer genes, despite an increasing publication bias for the inclusion of the term cancer in biomedical publications. Our studies have also shown that the distinctive dynamics of publication metrics of cancer genes and bystander genes help the identification of cancer genes that are not yet listed among the genes of the Cancer Gene Census [17].

## 2. Materials and Methods

### 2.1. Datasets Analyzed

PubMed_Timeline_Results_by_Year.csv files for human genes were downloaded from the PubMed website (https://pubmed.ncbi.nlm.nih.gov/) on 10 March 2022. We have also analyzed the PubMed dataset included as Appendix A of the paper of de Magalhães [1].

### 2.2. Lists of Cancer Genes and Bystander Genes

As the gold standard of ‘known’ cancer driver genes, we have used the lists of 54 oncogenes and 71 tumor suppressor genes identified by Vogelstein et al., (V-genes) [10]. As another list of known cancer driver genes, we have also used the 719 cancer genes of the Cancer Gene Census (CGC-genes) [17]. Information on the involvement of CGC-genes in gene-fusion events was obtained from the gene fusion mutation dataset downloaded from the COSMIC database (https://cancer.sanger.ac.uk/cosmic/, accessed on 12 February 2022).

Lists of bystander/passenger genes, candidate oncogenes, tumor suppressor genes and tumor essential genes defined in our earlier work were from Supplementary files S2, S4, S5 and S30 of that study [3].

### 2.3. Analyses of Publication Dynamics of Genes in the Context of Cancer Research

Changes in the proportion of PubMed publications on genes that mention cancer were monitored for the period of 1970–2021. Time points with less than 10 publications on the given gene were excluded from the graphical representations of publication dynamics.

### 2.4. Statistical Analyses

The statistical package of Origin 2018 was used for data processing and statistical analysis. Statistical significance was set as a *p* value of <0.05.

### 2.5. Network Analyses

Gene co-occurrence (or connectivity) graphs of CGC fusion genes were defined by the vertex set consisting of all genes present in the gene fusion mutation dataset of the COSMIC database. Two genes (vertices in the network) were regarded as being connected by an edge in the network if they co-occur in at least one fusion gene. Pajek 32-XXL 5.14, a program for large-network analysis and visualization [18]; (http://mrvar.fdv.uni-lj.si/pajek/, accessed on 12 February 2022) was used for the calculation and visualization of gene-fusion networks.

## 3. Results and Discussion

We have first analyzed the dataset of de Magalhães that lists the number of publications associated with human genes in PubMed (termed All_Pubs) and those publications on the genes that also mention cancer (termed Cancer_Pubs) [1]. Assuming that the number of publications associated with a given human gene (All_Pubs) reflects the intensity of research on that gene, whereas the number of publications that also mention cancer (Cancer_Pubs) reflects the intensity of work on the role of that gene in the context of cancer research, one expects that the numbers in Cancer_Pubs would be the highest for known cancer genes. Surprisingly, the top-ranking genes in Cancer_Pubs are not particularly enriched in known cancer genes (see Appendix A). Out of the 100 top-ranking genes, only 22 come from the cancer driver gene list of V-genes [10] and only 42 come from the more generous cancer gene list of the Cancer Gene Census (CGC-genes) [17].

A survey of the literature on the 100 top-ranking genes in Cancer_Pubs, revealed that, for many of the “non-cancer” genes (defined as genes absent in the CGC gene set), there is no evidence for their role in carcinogenesis. We have noted that “non-cancer” genes with the highest numbers in Cancer_Pubs, such as genes *TH*, *PC*, *IMPACT* and *ACHE*, have significantly higher numbers in All_Pubs than in Cancer_Pubs (Appendix A), reflecting the fact that research on these genes has not focused on cancer. Conversely, in the case of known cancer genes (V-genes) with the highest values in Cancer_Pubs (e.g., *EGFR*, *ERBB2*, *TP53*, *BRCA1*, *BRAF*, *KRAS*), the numbers in Cancer_Pubs and in All_Pubs are comparable, suggesting that the Cancer_Pubs/All_Pubs ratio (reflecting focus of research on cancer) might be a better indicator of cancer genes.

To test the suitability of Cancer_Pubs/All_Pubs ratios for the distinction of cancer genes and bystander genes, we have analyzed different subsets of the dataset of Appendix A; to increase statistical power, we have used increasing publication cut-off values (0, 10, 100, 500, 1000, 5000, 10,000 publications) in All_Pubs (Appendix A). Our analyses have confirmed that for the best-known cancer genes (e.g., *BRCA2*, *KRAS*, *PIK3CA*, *ERBB2*, *BRAF*, *IDH1*, *NRAS*, *BRCA1* and *ABL1*), the Cancer_Pubs/All_Pubs ratio is very high, close to 1.0 (see for example Appendix A, cut-off 1000).

In the case of the V-genes, the average value of Cancer_Pubs/All_Pubs was 0.65829 ± 0.21241 (at cut-off 0), a value that is significantly (*p* < 0.05) higher than the value of 0.32873 ± 0.24935 calculated for the passenger genes present in the dataset (Appendix A). The average value of Cancer_Pubs/All_Pubs of 0.55428 ± 0.22967 for the CGC-genes present in the dataset (at cut-off 0) was also significantly higher (*p* < 0.05) than that calculated for passenger genes (Appendix A). As shown in Appendix A, these differences between passenger genes and cancer genes were also significant at higher publication cut-off values: at cut-off 10000, the average values of Cancer_Pubs/All_Pubs for V-genes (0.6814 ± 0.30702) and CGC-genes (0.55809 ± 0.28948) were significantly (*p* < 0.05) different from those of passenger genes (0.25511 ± 0.20613).

As a corollary, the 100 top-ranking genes with the highest Cancer_Pubs/All_Pubs values are enriched in known cancer genes (Appendix A). For example, in the case of genes with more than 1000 papers in All_Pubs, the 100 genes with the highest Cancer_Pubs/All_Pubs values contained 65 genes from the list of CGC-genes, 36 of which were also present in the list of V-genes (Appendix A). These observations suggest that the Cancer_Pubs/All_Pubs ratios may help the distinction of cancer genes and bystander genes and thus may promote the identification of cancer genes. Nevertheless, it appears that the Cancer_Pubs/All_Pubs ratio per se does not necessarily reflect the importance of a gene for cancer. First, several known cancer genes of the V-gene list have rather low Cancer_Pubs/All_Pubs ratios (Appendix A). For example, *AR*, the gene for the androgen receptor has a Cancer_Pubs/All_Pubs value of 0.19197. Our analyses, however, confirmed that this low ratio is primarily due to misidentification of papers on the *AR* gene. When we carried out PubMed searches ensuring disambiguation of *AR* (including the terms AR and ‘androgen’), the Cancer_Pubs/All_Pubs value for the AR androgen was increased to 0.61395 (Appendix A). Thus, the case of the androgen receptor gene *AR* illustrates the point that it is essential to guarantee that papers relevant for the selected genes are correctly identified. To avoid misidentification, we have used terms for disambiguation; disambiguating terms for gene names were taken from UniProt (https://www.uniprot.org/, accessed on 12 February 2022). In addition to the *AR* gene, disambiguation had significantly increased the Cancer_Pubs/All_Pubs values for the *CIC*, *MET*, *APC*, *KIT*, *ATM*, *CBL*, *RB1* and *MPL* genes (Appendix A). Our analyses have shown that disambiguation is especially critical in cases where the gene names are not unique enough to avoid confusion with other terms. For example, we have noted that the unexpectedly high numbers of publications for genes *TH*, *PC*, *IMPACT* and *ACHE* in the dataset of de Magalhães (Appendix A) reflects such errors in data retrieval, rather than intense research on these genes (data not shown).

Another important message of the analysis of publications on the androgen receptor gene *AR* is that it is important to follow the temporal dynamics of the Cancer_Pubs/All_Pubs ratio, as it may reach high values for cancer genes relatively late, as a result of research efforts. For example, in the case of the *AR* gene, the Cancer_Pubs/All_Pubs ratio for the disambiguated publications showed a linear increase between 1995–2021, from 0.25 to 0.7 (see Figure 1A, Appendix A). (Note that, in the absence of disambiguation, there was just a very slow increase of the Cancer_Pubs/All_Pubs parameter (Figure 1A, Appendix A).

The time-dependent increase of Cancer_Pubs/All_Pubs ratios can also be illustrated with the cancer gene *BRAF*, where the number and proportion of papers on cancer started to rise only in 2002, elevating Cancer_Pubs/All_Pubs values from ~0.3 to ~0.91 (Figure 1B, Appendix A). A similar pattern, reflecting the impact of accumulating evidence on Cancer_Pubs/All_Pubs values, is seen in the case of cancer genes *FLT3* (Figure 1C), *U2AF1*, *ATRX*, *MED12*, *STAG2*, *FUBP1*, *SF3B1*, *GNA11*, *SPOP*, *BRAF*, *IDH1*, *BAP1* and *IDH2* (Appendix A). As a result of the time-dependent increase of Cancer_Pubs/All_Pubs ratios, the values calculated for the entire publication history of genes (CAt) may be different from those achieved as a result of a relatively late scientific break-through (CA2017–2021, the average values calculated for years 2017–2021, see Appendix A). For example, the CAt and CA2017–2021 values of *FLT3* are 0.72562 and 0.84559, respectively (Appendix A). Nevertheless, despite the influence of publication dynamics, the CAt values for V-genes are not significantly different from their CA2017–2021 values (Appendix A) since the discoveries relevant for the cancer gene status of a gene lead to a marked increase in the number of publications discussing their role in cancer.

In the case of genes that were recognized as cancer genes very early, such as *TP53* (Figure 1D), *HRAS*, *BRCA2*, *KRAS*, *ERBB2* (Figure 1E), *NRAS* and *NF1* the CAt and CA2017–2021, values are usually very similar (Appendix A). For example, the CAt and CA2017–2021 values of *TP53* are 0.87849 and 0.88056, respectively (Appendix A).

Analysis of publications on *TSHR*, the gene for the thyrotropin receptor cautions that even well-known cancer genes do not necessarily achieve high Cancer_Pubs/All_Pubs values, if research on that gene focuses on questions distinct from cancer (e.g., another major disease). The Cancer_Pubs/All_Pubs ratio for this gene is very low (0.20446) and it has not changed over the period from 1990–2021, illustrating that research on this gene is not limited to its role in carcinogenesis (Figure 1F). In fact, a higher fraction (0.5375) of publications on *TSHR* deal with hyperthyroidism or hypothyroidism than with cancer; this explains why *TSHR* can’t reach very high Cancer_Pubs/All_Pubs values.

Next, we have analyzed the publication dynamics of a representative set of passenger genes (randomly selected from the dataset, All_Pubs, cut-off value 1000, Appendix A), with a view of determining the influence of publication bias on Cancer_Pubs/All_Pubs values. Our analyses have shown that there is a general tendency for a gradual, yearly increase in Cancer_Pubs/All_Pubs ratio for passenger genes, such as *CYP2C19*, *VWF*, *NOX4*, *TLR4* and *LRRK2* (Appendix A, Figure 2). This linear increase reflects the bias favoring inclusion of the term “cancer” in publications in life sciences, as illustrated by the fact, that similar tendencies are observed for the association of the term “cancer” with neutral terms such as “protein”, “cell” or “tissue” (Appendix A, Figure 2F). The list of passenger genes includes genes for such intensely studied proteins as VWF (von Willebrand factor), C3 (Complement C3) and APOB (Apolipoprotein B-100), which have never been implicated in carcinogenesis, underlining the fact that in these cases, the increase in Cancer_Pubs/All_Pubs ratio simply reflects bias favoring the term ‘cancer’. Nevertheless, this publication bias has limited influence on the Cancer_Pubs/All_Pubs ratio of genes: the CA2017–2021 values of the passenger genes do not exceed 0.2 (Appendix A, Figure 2). This observation confirms that the Cancer_Pubs/All_Pubs ratios may help the distinction of cancer genes and bystander genes and thus may promote the identification of cancer genes.

As a further test of this assumption, we have examined the publication metrics of genes that we have identified earlier as novel oncogenes, tumor suppressor genes and tumor essential genes [3]. Although in some cases the number of publications was too low (All_Pubs < 50), to permit meaningful analyses (*IDH3B*, Appendix A), the dynamics of Cancer_Pubs/All_Pubs ratios clearly support our conclusion that genes *AURKA* (Figure 3A), *YAP1* (Figure 3B), *SLC16A1* (Figure 3C), *TP73* (Figure 3D), *YES1* and *TTK* qualify as cancer genes (Appendix A). As discussed earlier [3], in the case of these genes, a large body of evidence has accumulated to support their key role in carcinogenesis as oncogenes (*AURKA*, *YAP1*, *YES1*), tumor suppressor genes (*TTK*, *SLC16A1*) or as tumor essential genes (*TP73*).

In the case of the tumor essential gene, *G6PD* (Figure 3E), the CAt ratio is very low (Appendix A). It should be noted, however, that the gene is involved in a major disease, (non-spherocytic hemolytic anemia, due to G6PD deficiency) and a large fraction of Pubmed publications on this gene (>0.6) are dedicated to this disease. Despite this limitation, it is noteworthy that the publication pattern of *G6PD* suggests that its importance for carcinogenesis is increasingly recognized in the last decade (Figure 3E).

In the case of *SLC2A1* (Figure 3F), the gene for glucose transporter GLUT1, there is a steeper yearly rise of Cancer_Pubs/All_Pubs value, indicating that there is a growing awareness of its importance in carcinogenesis (Appendix A). The relatively high CA2017–2021 value (0.56) is especially significant if we consider that a large portion of papers on this gene deal with neurological diseases (e.g., GLUT1 deficiency syndromes, epilepsy, dystonia, mental retardation) caused by mutations of this gene, indicating that there is an upper limit for the CA2017–2021 value, that is significantly lower than 1.0.

In summary, our studies indicate that publication dynamics of cancer genes and bystander genes have characteristic differences, and this may help the identification of cancer genes through analyses of dynamics of Cancer_Pubs/All_Pubs ratios for genes. As a further test of this assumption, we have analyzed the publication dynamics of the 100 top ranking genes, with the highest CAt values of Appendix A, All_Pubs cut-off 1000. The publication dynamics of 36 of these genes, present in the list of V-genes, have been discussed above (Appendix A, Figure 1); the results for 29 genes that are CGC-genes are shown in Appendix A. Analyses of the publication dynamics of these CGC-genes have shown that they have CAt and CA2017–2021 values that significantly exceed the value characteristics of passenger genes, consistent with their cancer gene status (Appendix A, Figure 4).

Surprisingly, this analysis has revealed that the gene *PRCC* (encoding the proline-rich protein PRCC of unknown function) with the highest CAt value (0.97704, Appendix A) is a false hit, due to significant contamination of Pubmed hits with mismatches for Papillary Renal Cell Carcinoma (PRCC). Disambiguation, focusing on papers that deal with the *PRCC* gene, has shown that there is practically no PubMed evidence to support the cancer gene status of this gene per se. The true “cancer” hits for this gene refer to it only as the most common fusion partner of TFE3 transcription factor. *TFE3* forms fusion-oncogenes with several other partners (e.g., *NONO* in renal cell carcinoma, *ASPSCR1* in alveolar soft part sarcoma, *CLTC* in renal cell carcinoma and *SFPQ* in perivascular epithelioid cell tumor), suggesting that *TFE3* plays a more important role in carcinogenesis than its various ancillary fusion partners (Figure 5). Our observation is that there is no CAt based evidence for the cancer gene status of the *PRCC* gene, whereas in the case of *TFE3*, the CAt value (0.80218) is very high, suggesting that this parameter may help the distinction of contributions of gene partners to fusion oncogenes. To check this possibility, we have examined the CAt parameters for CGC genes that are hubs of fusion networks versus those that are just partners of such hub genes. This analysis has confirmed that the genes that are central hubs with numerous partners in the fusion network have very high CAt values, whereas those that have few or no partners have lower CAt values (Appendix A, Figure 5). For example, analysis of the data (0 cut-off, All_Pubs) revealed that the CAt values of CGC genes with at least five fusion partners (0.75812 ± 0.20102) are significantly (*p* < 0.05) higher than those for CGC genes with only one fusion partner (0.5251 ± 0.24143). Thus, the example of the *PRCC* gene cautions that some CGC genes may have acquired their cancer gene status as fusion partners of cancer genes, rather than in their own right. It is well known that gene fusions may inactivate tumor suppressor genes or may activate proto-oncogenes by changing their regulatory properties and it is clear that in such cases the role of the genes that are fused may be non-equivalent. Whereas one of the partners is a proto-cancer gene, the other may be just a passenger gene.

As a final test of the utility of publication metrics for the discovery of cancer genes, we have examined whether the 35 genes in the top-ranking 100, with the highest CAt values that are not included in the lists of V-genes or CGC-genes also qualifying as cancer genes (Appendix A). Our analysis has identified three false hits due to ambiguity of gene names (*HCCS*, *GAN* and *SLN*); these gene names are confused with other terms (e.g., HCCS, for Hepatocellular carcinomas, HCCs, SLN for Sentinel Lymph Node, SLN). Following disambiguation, these genes had very low PubMed matches, precluding meaningful analyses. The remaining genes, however, all had publication dynamics and high CAt and CA2017–2021 values, suggesting that they are highly relevant to cancer (Appendix A, Figure 6).

We have surveyed the literature on these genes to explore the reasons why they are associated with cancer. Here, we summarize only the major conclusions of our analyses; for brief descriptions and key references on these genes, the reader should consult Appendix A. In our earlier work [3], we have summarized the evidence for three of these genes (*AURKA*, *YAP1*, *TWIST1*) as novel cancer genes; these summaries are not repeated here.

A survey of the literature on these genes has revealed that there are three major reasons for their relevance for cancer: they may play a key role in carcinogenesis, may be important as tumor markers and may serve as targets for tumor therapy. Our survey has revealed that the majority of these genes are bona fide cancer genes that contribute to carcinogenesis, but in the case of some other genes, known as tumor markers, genes relevant for tumor therapy, it is less clear whether they play a critical role in carcinogenesis.

Genes for tumor markers include *MLANA* that encodes melanoma antigen recognized by T-cells, *MKI67*, the gene for proliferation marker protein Ki-67, *ALDH1A1*, the gene for aldehyde dehydrogenase 1A1, a marker of stem cells and *CD7*, the gene for T-cell antigen CD7, a tumor marker for acute myeloblastic leukemia (AML) and acute lymphoblastic leukemia (ALL). Genes relevant for tumor therapy include *ERCC1*, the gene for DNA excision repair protein ERCC-1 implicated in Cysplatin resistance and *ABCG2*, the gene for broad substrate specificity ATP-binding cassette transporter ABCG2 that plays a role in multidrug resistance.

The remaining genes appear to be cancer genes; based on a survey of the literature, they could be assigned to various cellular processes of cancer hallmarks in which they are involved (Table 1). For example, the genes for DNA repair protein XRCC1 (*XRCC1*) and E3 ubiquitin-protein ligase XIAP (*XIAP*) are involved in the hallmark of genome and proteome maintenance, the TGFβ subfamily member Nodal homolog (*NODAL*) plays a role in sustained proliferation and the reprogramming of metabolism of tumor cells, and programmed cell death protein 1 (*PDCD1*) is involved in the evasion of immune destruction of tumor cells, whereas vascular endothelial growth factor C (*VEGFC*) promotes tumor progression through the induction of angiogenesis.

It is noteworthy that some of the cancer genes identified in the present work as genes with very high Cat values have been identified earlier as members of the candidate cancer gene set, showing signs of strong positive selection for driver mutations in their coding region; see for example *AURKA* and *YAP1* [3]. The *PDCD1* gene identified in the present study as a gene with high CAt value is also present in the gene set displaying strong signs of positive selection for driver mutations [3], providing an additional argument for the cancer gene status of this gene. Interestingly, the publication dynamics of the *PDCD1* gene indicates that its involvement in carcinogenesis has been recognized only recently (Figure 6C). There are two additional genes, *XIAP* (Figure 6D) and *FOLH1* (Figure 6E), that also show significant signs of positive selection for missense mutation; their mutation parameters deviate by more than 1SD from those for bystander genes; see Supplementary file S5 of [3]. Furthermore, a recent study has identified genes *PGR* and *E2F1* as tumor suppressor genes under significant selection in tumors [4].

However, the majority of the cancer genes with high CAt values that did not show significant signs of positive or negative selection of their coding region during tumor evolution [3]. A possible explanation for this apparent contradiction between the proposed cancer gene status of these genes and lack of selection of their coding region is that selection may act on non-coding regions that control the expression of these genes. A typical example of such driver genes is *TERT*, the gene for the telomerase reverse transcriptase. The coding-region of this gene shows no significant sign of selection [3], whereas its promoter is a target for driver mutations in many types of cancer [19,20]. It is noteworthy in this respect that recent studies have shown that the enhancer region of one of the cancer genes identified here, *TNFSF10*, is subject to somatic mutation in kidney cancer [20].

An alternative explanation for the lack of selection of the coding region of some cancer genes is that these genes belong to the group of Epi-driver genes, rather than Mut-driver genes [10], i.e., they are expressed aberrantly in tumors due to aberrant promoter methylation. It is noteworthy in this respect that the promoter hypermethylation has been shown to inactivate two of the cancer genes, *CDKN2B* and *RASSF1A*, that appear to function as tumor suppressor genes in various types of cancer [21,22,23,24].

## 4. Conclusions

We have shown that known cancer genes (V-genes and CGC-genes) have significantly higher Cancer_Pubs/All_Pubs (CAt and CA2017–2021) ratios than bystander genes, suggesting that these parameters of research metrics may help the identification of cancer genes.

We have also shown that, although there is significant and increasing publication and funding bias favoring the inclusion of the term “cancer” in biomedical publications, this bias does not increase Cancer_Pubs/All_Pubs ratios to prevent the distinction of cancer genes and bystander genes. Paradoxically, this bias increases the reliability of the distinction of bystander genes and cancer genes, since the eagerness of scientists to prove the relevance of a gene for cancer research weakens the argument that if a gene has not been associated with cancer yet, it just means that its role in cancer has not been studied enough. One may argue that if a gene has been intensely studied but the CA parameters remain very low, it is very unlikely that future research will identify it as an important cancer gene.

In harmony with the expectation that high CA values may be used for the identification of cancer genes, our survey of the literature on genes with high CAt values, but not previously assigned to the cancer gene category, has shown that the majority of these genes qualifies as cancer genes, involved in cellular processes contributing to carcinogenesis (Appendix A, Table 1).

## Figures and Tables

**Figure 1 genes-13-01105-f001:**
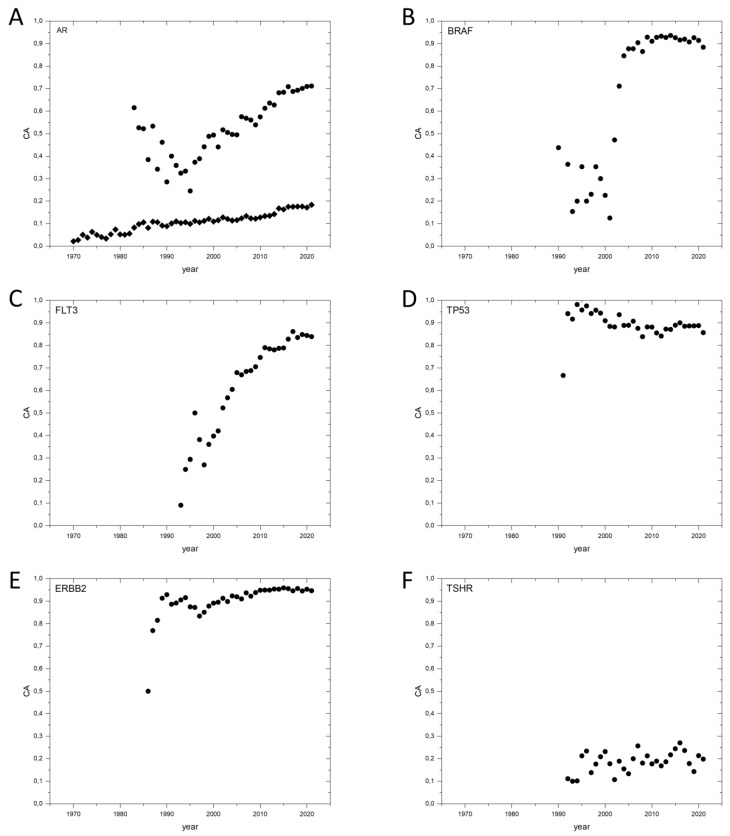
Dynamics of CA (Cancer_Pubs/All_Pubs) parameters of selected human cancer genes defined by Vogelstein et al., 2013 [10] (V-genes). (**A**): *AR*, androgen receptor. Diamonds represent the data for publications identified by the gene name “AR”; filled circles represent the AR data disambiguated with the term “androgen”. (**B**): *BRAF*, Serine/threonine-protein kinase B-raf. (**C**): *FLT3*, Receptor-type tyrosine-protein kinase FLT3. (**D**): *TP53*, Cellular tumor antigen p53. (**E**): *ERBB2*, Receptor tyrosine-protein kinase erbB-2. (**F**): *TSHR*, Thyrotropin receptor.

**Figure 2 genes-13-01105-f002:**
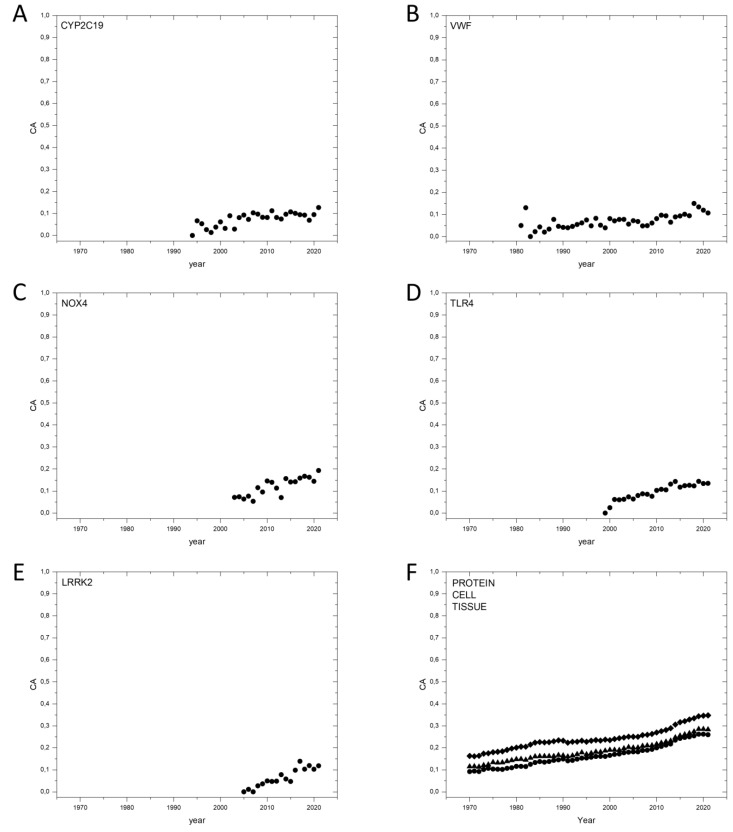
Dynamics of CA (Cancer_Pubs/All_Pubs) parameters of selected human bystander genes defined by Bányai et al., 2021 [3]. (**A**): *CYP2C19*, Cytochrome P450 2C19. (**B**): *VWF*, von Willebrand factor. (**C**): *NOX4*, NADPH oxidase 4. (**D**): *TLR4*, Toll-like receptor 4. (**E**): *LRRK2*, Leucine-rich repeat serine/threonine-protein kinase 2. (**F**): terms: “protein” (black circle); “cell” (black diamond); “tissue” (black triangle).

**Figure 3 genes-13-01105-f003:**
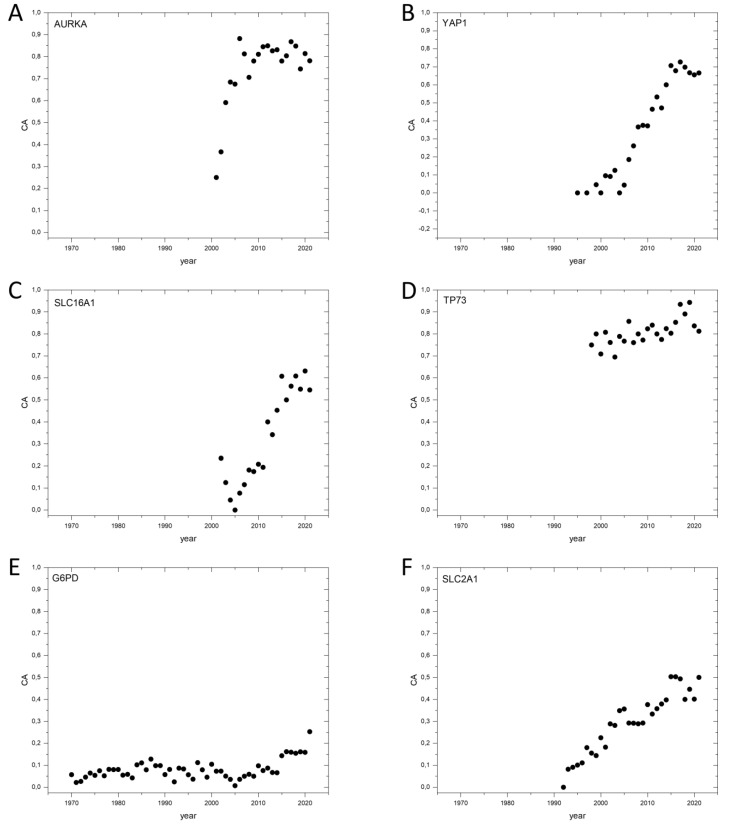
Dynamics of CA (Cancer_Pubs/All_Pubs) parameters of selected novel human cancer genes identified by Bányai et al., 2021 [3]. (**A**): *AURKA*, Aurora kinase A. (**B**): *YAP1*, Transcriptional coactivator YAP1. (**C**): *SLC16A1*, Monocarboxylate transporter 1. (**D**): *TP73*, Tumor protein p73. (**E**): *G6PD*, Glucose-6-phosphate 1-dehydrogenase. (**F**): *SLC2A1*, Solute carrier family 2, facilitated glucose transporter member 1.

**Figure 4 genes-13-01105-f004:**
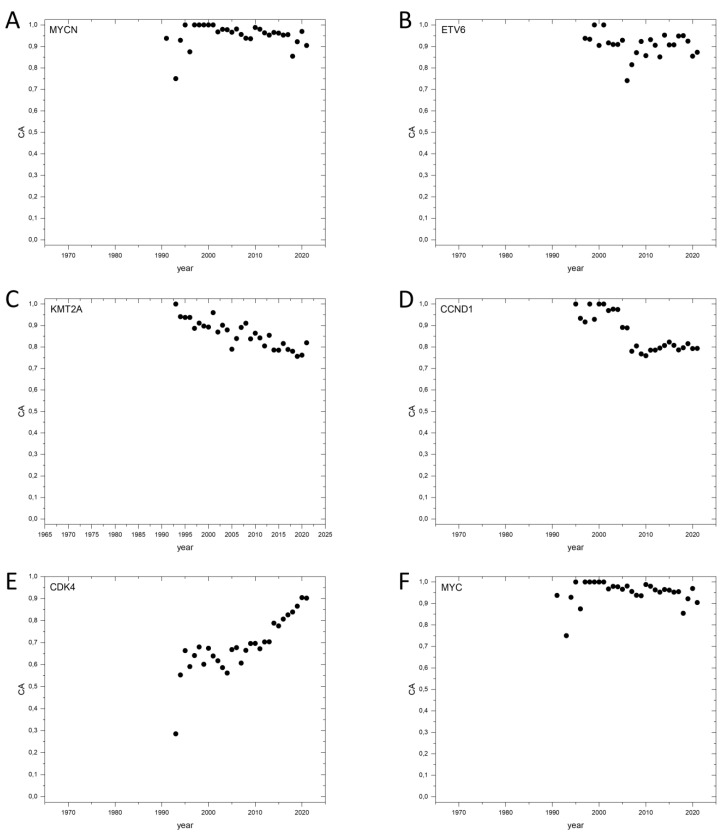
Dynamics of CA (Cancer_Pubs/All_Pubs) parameters of selected human cancer genes defined by Sondka et al., 2018 [17] (CGC-genes). (**A**): *MYCN*, N-myc proto-oncogene protein. (**B**): *ETV6*, G Transcription factor ETV6. (**C**): *KMT2A*, Histone-lysine N-methyltransferase 2A. (**D**): *CCND1*, G1/S-specific cyclin-D1. (**E**): *CDK4*, Cyclin-dependent kinase 4. (**F**): *MYC*, Myc proto-oncogene protein.

**Figure 5 genes-13-01105-f005:**
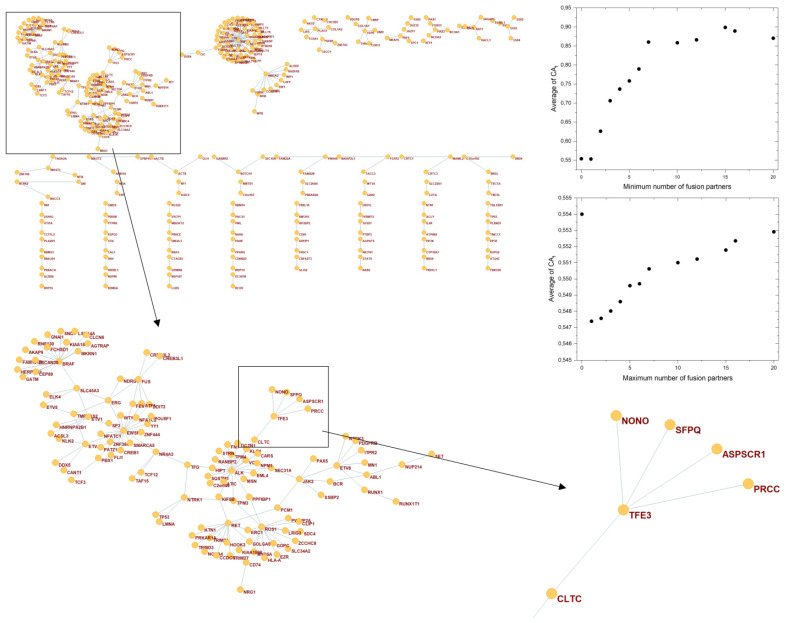
CA_t_ (Cancer_Pubs/All_Pubs values calculated for the entire publication history) of genes in the fusion network of human cancer genes, defined by Sondka et al., 2018 [17] (CGC-genes). The figure shows the fusion network of CGC-genes with at least one fusion partner, highlighting the fusion partners of the *TFE3* gene. The insets show the dependence of average CA_t_ values of CGC genes on the number of fusion partners at All_Pubs, cut-off 0.

**Figure 6 genes-13-01105-f006:**
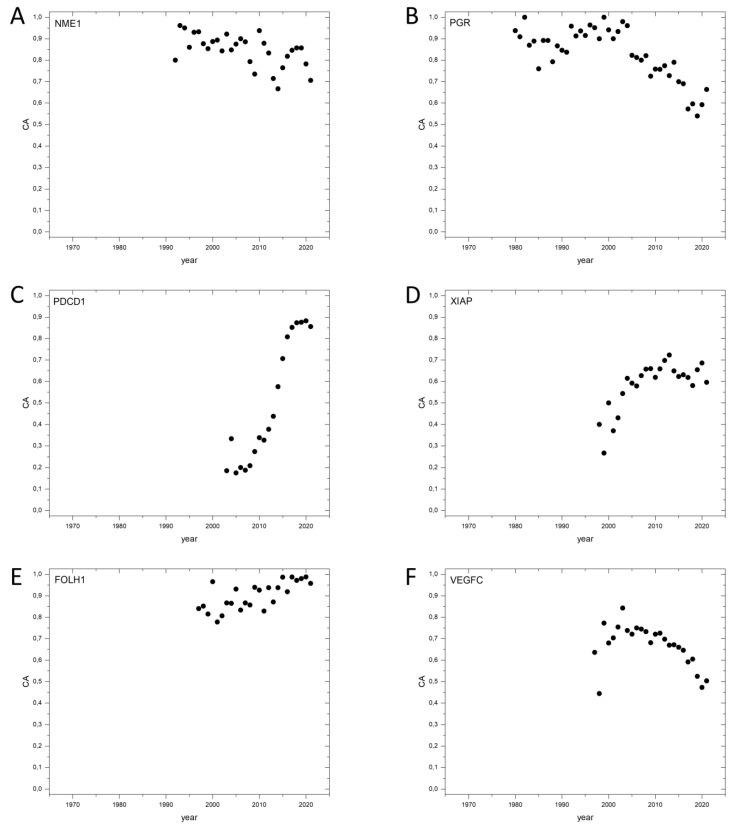
Dynamics of CA (Cancer_Pubs/All_Pubs) parameters of selected genes with high CA_t_ values but missing from the lists of known cancer genes. (**A**): *NME1*, Nucleoside diphosphate kinase A. (**B**): *PGR*, Progesterone receptor. (**C**): *PDCD1*, Programmed cell death protein 1. (**D**): *XIAP*, E3 ubiquitin-protein ligase XIAP. (**E**): FOLH1, Glutamate carboxypeptidase 2. (**F**): *VEGFC*, Vascular endothelial growth factor C.

**Table 1 genes-13-01105-t001:** Assignment of cancer genes with high CA_t_ values to key cellular processes of carcinogenesis.

Hallmarks of Cancer	Gene Symbol
Defects of genome, epigenome, transcriptome or proteome maintenance	*TWIST1, FOXM1, SKP2, BMI1, PGR, XRCC1, XIAP*
Sustained proliferation	*AURKA, YAP1, FOXM1, CDKN2B, E2F1, CDC25C, NODAL*
Evasion of growth suppressors	*SKP2, BMI1*
Reprogramming of metabolism	*YAP1, CA9, FOXM1, NODAL*
Replicative immortality	
Evasion of cell death	*YAP1, RASSF1, BIRC5, BMI1, TNSF10, MCL1, E2F1, XIAP, NODAL*
Evasion of immune destruction	*SNAI1, PDCD1*
Tumor promoting inflammation	*FOXM1, TNSF10*
Inducing angiogenesis	*FOLH1, FOXM1, VEGFC*
Activation of invasion and metastasis	*FOLH1, NME1, CA9, CD99, SNAI1, FOXM1, BMI1, TNSF10, CD24, CCNB1, CDC25C, NODAL*

For annotation of genes identified in the present study, see Appendix A.

## Data Availability

All data generated or analyzed during this study are included in the manuscript and Appendix A.

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
