# Peer review of "Use of Publication Dynamics to Distinguish Cancer Genes and Bystander Genes"

_genes, 2022, doi:10.3390/genes13071105_

Round 1
Reviewer 1 Report
The primary goal of the work is to distinguish between bona fide and bystander genes that play role in cancer. Although the work is important, I think the authors have not done a comprehensive search of all the contributing factors to cancer. For example. there is no mention of any metal transport genes that are upregulated in cancer and chemoresistance. Copper transport proteins play a very important role in cancer and chemoresistance and are an example of the interaction of proteins with micronutrients to support tumor growth. There are many other examples like this that the authors have not looked into. I suggest extending their search into other areas like metal biology, immunotherapy, ROS etc.
Author Response
"Please see the attachment."

Reviewer 2 Report
In this paper, Bányai and colleagues use publication dynamics (frequency of publications containing a gene and the word cancer, amongst other permutations) to attempt to distinguish cancer related genes from bystander/passenger genes. Their work extends on that of de Magalhāes and colleagues and may explain some errors in data retrieval by de Magalhāes et al.
Cancer is a heterogeneous collection of diseases each driven by different molecular changes. Moreover, tumours from different patients with the same tumour type can exhibit different molecular drivers, as can different cells within the same tumour. Many of these molecular changes will be bystander/passenger events and so distinguishing them from driver events is important. Bányai and colleagues approach this in a novel way by using publication dynamics to identify genes recurrently described in the literature in the context of cancer.
This paper may be controversial in its approach and may itself prompt extensive discussion within the literature. I think this is a good thing and this work should be published to promote that discussion. However, I disagree with the conclusion that, "distinctive publication dynamics in cancer genes...may be used for the identification of novel cancer genes". The issue here is that, by definition, such genes have numerous publications identifying them as cancer-related genes and therefore cannot be described as novel cancer genes. In this reviewer's opinion, the approach to analysis of publication dynamics described by the authors would help in identifying which genes should be included in future research. This may include the design of targeted gene panels or virtual panels by research groups and/or biotechnology companies who invest a great deal of time and resources into choosing which genes to include in targeted panels and/or study designs. Their conclusions that this may help in the identification of novel cancer genes is mentioned several time throughout the manuscript and should be revised.
There is one limitation to the approach taken by Bányai and colleagues, which the authors openly describe in the Results and Discussion section. This limitation is that some gene require careful disambiguation from other terms, which can result in cacner-related genes with a low Cancer_Pubs/All_Pubs (CA) ration, as well as non-cancer-related genes with high CA ratios. It's hard to see how the output of their analysis can be trusted for all genes without delving deeply into each individual case. What is the workaround to ensure all genes are sufficiently disambiguated from other terms so that this is not an issue?
Figure legends should define CA and CAt where appropriate.
Figure 5 was illegible due to size/resolution issues and therefore difficult to follow
Author Response
"Please see the attachment."
